# From Rust to Quantum Biology: The Role of Iron in Retina Physiopathology

**DOI:** 10.3390/cells9030705

**Published:** 2020-03-13

**Authors:** Emilie Picard, Alejandra Daruich, Jenny Youale, Yves Courtois, Francine Behar-Cohen

**Affiliations:** 1Centre de Recherche des Cordeliers, INSERM, Sorbonne Université, USPC, Université Paris Descartes, Team 17, F-75006 Paris, France; adaruich.matet@gmail.com (A.D.); youale.j@hotmail.fr (J.Y.); courtois.yves@numericable.com (Y.C.); francine.behar@gmail.com (F.B.-C.); 2Ophthalmology Department, Necker-Enfants Malades University Hospital, APHP, 75015 Paris, France; 3Ophtalmopole, Cochin Hospital, AP-HP, Assistance Publique Hôpitaux de Paris, 24 rue du Faubourg Saint-Jacques, 75014 Paris, France

**Keywords:** iron, retina, transferrin

## Abstract

Iron is essential for cell survival and function. It is a transition metal, that could change its oxidation state from Fe^2+^ to Fe^3+^ involving an electron transfer, the key of vital functions but also organ dysfunctions. The goal of this review is to illustrate the primordial role of iron and local iron homeostasis in retinal physiology and vision, as well as the pathological consequences of iron excess in animal models of retinal degeneration and in human retinal diseases. We summarize evidence of the potential therapeutic effect of iron chelation in retinal diseases and especially the interest of transferrin, a ubiquitous endogenous iron-binding protein, having the ability to treat or delay degenerative retinal diseases.

## 1. Introduction

Iron is a major element in biology. Besides its well-known role in prebiotic conditions after the rise of oxygen in the atmosphere, its insolubility led to the development of many mechanisms to allow the primitive cells and organisms to use it. They are driven by the transition of ferrous iron (Fe^2+^) to ferric iron (Fe^3+^) involving an electron which is particularly available and is the basis of vital functions and dysfunctions in the organs.

In this review, we analyze several of the well-known or recently discovered functions of iron in the eye, mainly in the retina, and the most promising approaches to regulate it and improve a large number of its negative side effects which can lead to vision impairment. We will focus on the main functions of transferrin (TF) as a partner in the systemic and cellular mechanisms that underlie the regulation of iron homeostasis and its disorders.

## 2. The Retina Structure and Retinal Oxygen Supply

The eye is a complex and confined organ formed by different compartments and structures essential for the transmission and focus of photons from the cornea to the photoreceptors (PRs), which convert them into an electrical signal transmitted to the brain. The neural retina comprises the PRs, cones, rods, the interneurons, the ganglion cells, the glial cells such as retinal Müller’s glial cells (MGC), astrocytes, and microglia (Figure 1). The retina is vascularized by two separate vascular systems, the retinal vessels, branches of the central retinal artery that vascularize the inner retinal layers, and the choroidal vessels, branches of the ciliary arteries that supply the avascular PR layer through the retinal pigment epithelium (RPE) cells. In primates and human, visual acuity, photopic vision, and color vision are ensured by the macula, a highly specialized retinal area that comprises less than 5% of the total retinal surface, located at the center of the visual axis. The center of the macula, the fovea, is devoid of retinal vessels and composed exclusively of cones and MGC cells.

The retina is separated from the circulation by two blood-retinal barriers (BRB). The inner BRB consists of a neuro-glio-vascular complex formed by tight junctions between endothelial cells of the retinal capillaries, pericytes, astrocytes, MGC, and microglia [1]. The outer BRB is formed by the tight-junction monolayer of RPE cells that are in close contact with the choriocapillaries, which control exchanges through diaphragmed fenestrations [1]. Oxygen is the most supply-limited metabolite in the retina [2]. Its supply to the retina is ensured by the choroid, which provides oxygen to the outer retina, whilst the retinal circulation provides the oxygen requirements of the inner retina. In normal condition, the level of oxygen tension (Po_2_) in the outer retina is ten times lower than in the inner retina [3]. Oxygen and glucose consumption are metabolized to lactate, while aerobic glycolysis dominates energy production in the outer retina. Several factors modify Po_2_ level and utilization at the cellular level: the retinal depth, the light, and hyperoxia [4,5]. PRs have almost all mitochondria in their inner segments far from blood vessels. Light decreases oxygen utilization on the outer retina as much as by a factor of two and increase Po_2_. Hyperoxia dramatically increases Po_2_ in the retina with the increase higher in outer retina compared to inner retina. The development and maintenance of retinal vasculature is regulated by α subunits of hypoxia-inducible factor (HIF), which induce genes required for retinal homeostasis, such as vascular endothelial growth factor (VEGF) under hypoxic conditions [6]. HIF proteins, which act as regulators of oxygen homeostasis, also depend on iron for their activity, and they regulate genes involved in iron metabolism [7]. Hyperoxia is deleterious to the outer retina as the oxygen leads to the formation of reactive oxygen species (ROS) according to the Fenton and Haber–Weiss reaction catalyzed by iron to generate RO° radical (review in [3]). Iron and oxygen are thus closely linked in retinal metabolism in health and disease conditions.

## 3. Iron Homeostasis in the Retina

### 3.1. Distribution of Iron in the Retina

Iron is widely and unevenly distributed throughout the adult rat retina. The highest concentrations of iron were observed by proton-induced X-ray emission in the choroid, the RPE, and the inner segments of the PR. PR outer segments also contain iron, as inclusions inside the discs [8]. Iron and iron-related parameter (total iron binding capacity, TF and TF saturation percentage) distribution in the eye are different between diurnal and nocturnal animals. In cow and pig retina, iron concentration is higher than in rat retina, suggesting that the nocturnal habit of living could influence iron-related parameters in the retina [9]. The iron level also varies during retinal development and aging. Moos et al. have shown that in rats, iron entry is very high during retinal development and maturation, then decreases in adulthood, and increases again with aging [10]. In rodents, there are gender and strain-specific influences on iron regulation in the neural retina [11]. Human sex-associated differences in iron levels have also been reported, women having more retinal iron than men at all ages [12]. With aging, iron deposits are found in the RPE/choroid complex in rats and in the stroma of the choroid in non-human primate regardless of serum iron concentration [13]. Increased iron levels in the retina have also been reported in human eyes with age [12]. In rodent eyes, both neural retina and RPE/choroid present an increase of iron concentration which is associated with modifications of iron-related proteins mRNA and protein levels [14,15].

### 3.2. Proteins Involved in Retinal Iron Homeostasis

#### 3.2.1. General Iron Homeostasis

Under physiological conditions, almost all non-heme iron (Fe^3+^) in the circulation is transported bound to TF (transferrin-bound iron: TBI) with high affinity (for review on cellular iron metabolism [16]). At cellular level, the TF with two Fe^3+^ (holo-TF) is bound by its receptor (TFR1), and the complex is internalized. The Fe^3+^ is released from TF in the endosome under the effect of acidification through the action of an ATP-dependent proton pump. Iron is then reduced into the ferrous form, Fe^2+^, by endosomal ferrireductase six transmembrane epithelial antigen of the prostate 3 (STEAP3) and exported by divalent metal transporter 1 (DMT1) in the cytosol where it contributes to the labile iron pool (LIP). In case of iron overload, TF is saturated with iron, and the non-transferrin-bound ferrous iron (NTBI) could be up-taken by iron importers such as the ZRT/IRT-like proteins (ZIPs) or DMT1 and joined intracellular LIP. The LIP consists of a transitory pool of iron species associated with a variety of ligands with low affinity (citrate, phosphate, inorganics irons) and easily oxidized, in transit to be distributed to the organelles (in particularly in mitochondria or nucleus) for cell metabolism in iron requiring proteins, stored or released.

In non-erythroid cells, the majority of iron is stored in ferritin (FT). Composed of 24 subunits of both heavy (HFT) and light (LFT) chains, FT forms a tissue-specific heterocomplex which can store up to 4500 Fe^3+^. Fe^2+^ from LIP is transported to FT by Poly(rC)-binding proteins (PCBP1 and PCBP2) and is oxidized by ferroxidase activity of HFT in an oxygen-dependent manner. Then Fe^3+^ is stored in the cavity formed by LFT. Iron is released from FT in a controlled manner by autophagy involving a cargo receptor, the nuclear receptor coactivator 4 (NCOA4).

The only known mammalian iron exporter is ferroportin (FPN), a transmembrane protein which requires a multicopper ferroxidase to convert Fe^2+^ to Fe^3+^, allowing binding to TF. Hephaestin (HEPC), ceruloplasmin (CP), amyloid-beta precursor protein (APP), and the newly identified zyloklopen (ZP) are co-localized with FPN at the surface membrane or secreted.

#### 3.2.2. Iron Flux in Retina

A local retinal homeostasis of iron, independent from the systemic regulation, is suspected by the fact that main proteins involved in iron homeostasis, previously confined to systemic expression, are locally synthesized in the retina (Table 1). In addition, the outer and inner BRB prevents the entry of large quantities of iron into the eye in case of systemic iron overload. The study of mouse models invalidated for iron homeostasis proteins (Table 1) led to a hypothetical model in which iron entry and homeostasis in the retina is divided into two compartments, carried by the RPE and MGC, delimited by the external limiting membrane, and having limited exchanges in physiological condition. 

##### Transferrin-Bound Iron Transport in the Retina

The RPE imports iron bound to TF from the choriocapillaries through the transcytosis of TFR1 present at the basal membrane of the RPE (Figure 2). The transcytosis of the TF/TFR1 complex along microtubules via galectin 4 and Rab11a [42] has been described in vitro. The presence of TFR1 on the apical side of RPE is ambiguous and suggests that TF/TFR1 transcytosis or a potential iron-TF uptake by RPE could egress iron from the outer retina to choriocapillaries. Six hours after an intravitreal injection of holoTF tagged with a fluorochrome, TF is localized in RPE and choroid, which favors the later hypothesis [43]. Another iron entry in RPE is the phagocytosis of PR outer segments which contains high quantities of iron [8]. Once in RPE cytosol, iron is stored in FT and in melanosomes [44]. The release of iron from cell is possible through FPN present at the basal membrane of RPE and a multicopper ferroxidase. HEPH, CP, and APP but not ZP are expressed in RPE [26].

In the inner retinal layers, iron is imported through retinal endothelial cells (REC) which express TFR1 at their luminal side [45]. Two mechanisms of iron transfer across the abluminal membrane of REC into the retina are evoked: TF/TFR1 transcytosis or/and TF/TFR1 endocytosis following by iron released from endosome and iron export by FPN. The abluminal membrane of REC expresses FPN colocalized with HEPC, CP, and APP [24,31]. Iron is bound by TF and distributed to the retina or the vitreous. With its unique position extending from the vitreous to PRs and its capacity to synthetize TF [46] and to express FPN [24], MGC plays a crucial role in the distribution of iron from the inner retina to the inner segment of the PR.

Iron presents in the non-vascularized subretinal space, between the apical side of RPE and the PR, is mainly bound to TF, secreted by PRs and RPE, and up-taken by PRs for their highly metabolism activities, by TFR1 express at inner segments. The PR inner segment is the iron storage compartment for PR segments, where both FT chains and mitochondrial FT are highly concentrated [30]. Iron export from the PR is ensured mostly by FPN also present in inner segments, CP and HEPH being poorly involved in favor of APP [26].

##### Non-Transferrin-Bound Iron Transport in the Retina

Whereas iron-bound TF is the main transport system to cross the BRB, transferrin-independent iron delivery to the retina can occur (Figure 1). Serum FT, exclusively composed of LFT has specific receptor, the scavenger receptor class A, member 5 (SCARA5) expressed in cytoplasm and nucleus in retinal endothelial cells, ganglion cells, astrocytes, the inner nuclear layer, MGC, microglia, outer nuclear layer, cones segments and RPE. After intravenous injection, serum FT remained confined in retinal endothelial cells in the inner retina [47]. Another protein possibly involved in iron transport is lactoferrin (LF), a multifunctional protein which shares 65% of homology with TF. It is synthesized in various human ocular tissues mainly in the RPE but not in the neural retina [18]. LF receptors have not been studied in the eye but are present in the brain [48]. Lipocalin 2 (LCN2) does not bind to iron directly, but through interaction with siderophores (catecholate and carboxylate) as cofactors [49] could also be implicated in iron transport in the retina [20]. β-hydroxybutyrate dehydrogenase-2 (Bdh2), an enzyme that is critical for the synthesis of 2,5-dihydroxybenzoic acid (2,5-DHBA), the mammalian siderophore, is found throughout the retina in all cell layers, including ganglion cells, MGC, and RPE cells [50]. Two major membrane-bound receptors for LCN2, megalin and 24p3R, have been identified in RPE [21,22]. Although LCN2 is being recognized as an important factor in retinal diseases [21], its exact contribution in iron retinal transport in health and diseases remain to be determined.

In case of iron overload, the NTBI could be up taken by iron importer ZIP or DMT1. This could explain why in retinal iron overload models, iron continues to accumulate despite the reduced expression of TFR1 in retina and RPE [31]. The specific localization of DMT1 in PRs and bipolar and horizontal cells suggests that it could be involved in providing iron to these cells, for phototransduction or neurotransmitter synthesis [30] ZIP8 and ZIP14 are expressed in RPE, choroid, REC, choroidal endothelial cells (CEC), ganglion cells, PRs, and MGC. At a high degree of TF saturation in the retina, there is a decreased ZIP14 expression whereas ZIP8 expression remains stable [31].

#### 3.2.3. Iron Regulation in Retina

Cellular iron uptake and release and the intracellular LIP size are tightly controlled. Transcriptional, post-transcriptional, and post-translational processes regulate iron homeostatic proteins (for a review, see [51]). The main mechanisms of intra- and extra-cellular regulation of iron levels are limited to two extremely controlled systems.

The first system includes iron regulatory proteins (IRP) 1 and 2—intracellular iron regulatory proteins which, depending on the amount of iron, bind iron responsive element (IRE) sequences present on the mRNAs of iron homeostasis proteins such as FPN, TFR1, FT, and DMT1. Depending on the position of the IRE site, IRP controls their translation or degradation. Under conditions of increased cellular iron, IRP1 loses its IRE-binding activity by acquiring an iron in the 4Fe–4S cluster, whereas IRP2, is degraded by proteasome. In this condition, *tfr1* and *dmt1* mRNA are degraded, whereas *ft*, *fpn*, and *hif-2α* mRNA are translated. The localization of IRP1 and IRP2 has not yet been identified in the retina but their expressions are ubiquitous in mammalian cells. Mice with *Irp1*^+/−^
*Irp2*^−/−^ genotype show more severe neurodegenerative disease than *Irp2*^−/−^ animals [30]. These IRP deficient retinas have increased FPN and FT in the inner segments, MGC endfeet, and inner retina compared to age and strain matched wild type retinas, suggesting that FPN and FT levels are regulated by IRPs in the retina [23]. In a model of light induced retinal degeneration, 2 h after light exposure, *Irp2* but not *Irp1* mRNA increased in the retina [32].

The second system focuses on hepcidin (HEPC), a peptide hormone principally synthetized by the liver. However, HEPC is also synthesized by PR, RPE, and MGC [39]. It is activated by two cellular signaling pathways induced by excess of iron, the transferrin receptor 2 (TFR2)/Human homeostatic iron regulator protein (HFE) pathway and the Bone Morphogenetic protein (BMP6)/Mothers against decapentaplegic homolog 1 (SMAD) pathway. When the TF saturation is high at the basolateral level of the RPE, the HFE is released from TFR1 and binds to TFR2, which activates the transcription of HEPC. BMP6 secreted by the retina and the RPE, binds to its receptors coupled to hemojuvelin (HJV) protein at the apical level of RPE in order to activate the synthesis of HEPC [35]. HEPC binds to the extracellular domain of FPN on the cell surface, leading to its internalization and degradation, effectively preventing cellular iron export and limiting the amount of iron that gets into the extracellular fluid. The specific deletion of HEPC in the retina does not lead to age-associated retinal iron accumulation, whereas liver-specific HEPC silencing leads to early serum, RPE, and retina iron accumulation followed by retinal degeneration [52].

Finally, the hypoxia inducible factor (HIF) acts as a transcription factor for certain iron homeostasis genes such as the *Tf*, *tfr1*, *Dmt1*, *Fpn*, and *Cp* genes by binding to a specific hypoxia-responsive element (HRE) site present on their mRNAs. Expression and degradation of HIF are also dependent on iron. In fact, Fe^2+^ is the cofactor of prolyl hydroxylase involved in the degradation of HIF-1α, and at the same time HIF-2α has an IRE sequence in the 5′UTR of its mRNA, which in the condition of iron deficiency, inhibits its translation. Nuclear staining of HIF-1α was observed in the GCL, the inner nuclear layer and the outer nuclear layer in human and rat [41]. Under retinal hypoxia, both HIF-1α and HIF-2α are activated but have cell specific expression within the inner retina. Specifically HIF-2α activation seems to play a key role in regulating the response of MGC to hypoxia [53].

## 4. Physiopathological Role of Iron in the Retina

### 4.1. Iron in Cellular Metabolism/Functions

#### 4.1.1. Iron as a Fe-S Structural Motif Involved in Various Cellular Machinery Proteins

Iron sulfur (Fe-S) proteins are characterized by the presence of Fe-S clusters localized in different cell compartments (for review [54]). IRP1 is a Fe-S cluster that participates in sensing and regulating iron homeostasis in the retina. Frataxin is a nuclear-encoded mitochondrial protein involved in Fe-S cluster assembly, heme synthesis, and intracellular iron homeostasis. Frataxin is an allosteric activator which binds to this assembly complex [55]. It is present in the retina [56] and in the RPE [57] and could be responsible for retinal neurodegeneration induced by defective mitochondrial function [58]. In addition, Fe-S clusters may act as biological sensors by their binding properties to molecular oxygen and nitric oxide [59] both critical for the retinal physiology and pathology.

#### 4.1.2. Iron in Nucleic Acids Machinery, Cell Proliferation, and DNA Repair

A recent review has reported the multiple implications of iron in DNA synthesis and repair, as well as in RNA metabolism [60]. Cytosolic and nuclear Fe-S proteins intervene in the genome stability [61]. Iron has been implicated in DNA synthesis and repair as a cofactor of sirtuin 2, an histone deacetylase, involved in iron homeostasis [62]. Sirtuin 2 maintains cellular iron levels by binding the nuclear factor erythroid-2-related factor 2 (NRF2) leading to a reduction in total and nuclear NRF2 levels. NRF2 is a transcription factor that plays key roles in retinal antioxidant and detoxification responses and has been linked with the development of age-related macular degeneration (AMD) [63].

Mitochondria are a major source of ROS and mitochondrial DNA is very susceptible to oxidative damage [64]. In RPE cells, mitochondrial DNA is damaged by hydrogen peroxide [65]. Deletions in mitochondrial DNA occurred in function of age in human neural retina [66], and the accumulation of age-related mitochondrial mutations in the eye has been correlated with a decrease in ATP production and increase ROS output, leading to oxidative stress, inflammation, and degradation [67].

#### 4.1.3. Iron in Oxygen Transport and Regulation

Hemoglobin is synthetized in the retina [68]. It is one of the main protein synthesized in primary cultures of human RPE and secreted in vivo through the basolateral membrane [69].

Under physiological condition, free hemoglobin is bound by haptoglobin, but in case of massive hemolysis, hemoglobin releases free heme which binds hemopexin. Both hemopexin and haptoglobin have been described in the human retina [70,71]. The mRNAs for both haptoglobin and hemopexin were detected in the neural retina and PR as well as ganglion cells but not in RPE cells.

Neuroglobin is a highly conserved oxygen-binding protein reviewed in [72] and highly expressed in the retina. Its role is to facilitate oxygen metabolism, being localized in mitochondria. Hemin, the ferric chloride salt of heme enhances neuroglobin expression and protects animal model of N-methyl-N-nitrosourea-induced retinal degeneration [73]. In this model, hemin protects also cones from apoptosis. Neuroglobin has also been associated with retinal damage induced by light [74] which may reflect the changes in iron metabolism first described with light on retina [32]. It has also been associated with VEGF expression and thus could participate in retinal angiogenesis [75].

Heme, Fe^2+^ protoporphyrin IX, the prosthetic group of hemoproteins including hemoglobin, neuroglobin, oxidases/peroxidases, or cytochromes can be released after auto-oxidation. Heme transporter proteins also intervene in iron metabolism in the retina, and their dysregulation could potentially cause oxidative cell damage. All three heme transporters feline leukemia virus subgroup C receptor (FLVCR), breast cancer resistance protein (BCRP), and proton-coupled folate transporter (PCFT/HCP-1) are expressed in the retina and RPE. In the RPE, the expression of FLVCR is restricted to the apical membrane and the expression of BCRP and PCFT to the basolateral membrane. In cases of iron overload, the expression of FLVCR and PCFT is upregulated and BCRP is downregulated, suggesting an important role of heme transporter proteins in retinal iron regulation [76].

#### 4.1.4. Iron and Visual Function

The involvement of iron in the vision cycle was discovered with the characterization of the enzyme RPE65, as an iron-dependent isomerohydrolase [77]. RPE65, abundant in the RPE [78], ensures the isomerization and hydrolysis of all-*trans* retinyl ester to 11-*cis* retinol. RPE65 is essential for vision, and mutations in *rpe65* genes induce Leber congenital amaurosis, a form of retinitis pigmentosa that leads to blindness [79]. Recently, RPE65 was also shown to intervene in the production of meso-zeaxantin, an ocular specific carotenoid which protects the fovea from oxidative stress [80].

An alternate pathway for 11-*cis* retinol recycling has been described in MGC by isomerases 1 or 2 that also appear to be iron dependent [81]. Few studies have analyzed the iron flux in the retina with the diurnal cycle conversely to what has been performed in brain in mice [82,83]. Among the sensory guanylate cyclase proteins and signaling network, guanylyl cyclase activating protein 5 is the only protein that binds strongly Fe^2+^ in zebrafish [84]. It is proposed as redox sensor in visual transduction.

Phototransduction depends on the phagocytosis of outer segments from PR by the RPE. The constant release of the outer segments from PR and their digestion during phagocytosis by RPE implies membrane biogenesis, a process which needs iron as a cofactor of fatty acid desaturase [85]. Royal College Surgeon (RCS) rats invalided for the phagocytosis protein Myeloid-epithelial-reproductive tyrosine kinase (MERTK) have increased iron in retina and particularly in RPE phagosomes and also increased retinal FT and TF expression [86].

Iron is also involved in neurotransmitters secretion as it regulates glutamate secretion by RPE cells via the cytosolic aconitase pathway [87]. Dopamine biosynthesis in specialized amacrine cells results from the conversion of the amino acid L-tyrosine in L-3,4-dihydroxyphenylalanine (L-DOPA) using oxygen and Fe^2+^ [88]. Synaptosomal nerve-associated protein 25 (SNAP-25) is a Fe-S protein involved in synapse vesicle fusion with plasma membranes highly present in retina [89].

A significant number of ATP binding cassette (ABC) transporters, involved in lipid trafficking in retinal cells, have been linked to severe genetic ocular diseases [90]. ABCA4 is present in the PR and transports 11-*cis* and all-*trans* isomers of *N*-retinylidene-phosphatidylethanolamine across disc membranes, preventing the accumulation of toxic bisretinoid lipofuscin compounds in PR and RPE cells. In *Abca4* null mutant mouse which presents accumulation of N-retinylidineN-ethanolamine (A2E) bisretinoids and lipofuscin in the RPE, intracellular iron accumulation is also observed which contributes to enhancing oxidative cell death [91]. The intracellular accumulation of iron in cells of the RPE in culture decreases the expression of the transporters of cholesterol ABCA1/ABCG1, increasing the level of pro-inflammatory cholesterol in retina [50].

### 4.2. The Dark Side of Iron

#### 4.2.1. The Crucial Role of Iron in Oxidative Stress-Mediated Damages in the Retina

The ability of iron to change easily its valence and switch between the Fe^2+^ and Fe^3+^ forms, providing or accepting electrons, respectively, ensures a privileged position in living matter as mediator of key biochemical reactions. However, the presence of free labile iron in cell or NTBI in circulation is prone to generate highly ROS in the Fenton/Haber–Weiss reaction.

$Fe^{2+}+H_{2}O_{2\;}\;\to Fe^{3+}+OH^{-}+HO^{\cdot }$: Fenton reaction

$O_{2}{}^{\cdot -}+Fe^{3+}\to O_{2}+Fe^{2+}$: Haber–Weiss reaction

$O_{2}{}^{\cdot -}+H_{2}O_{2\;}\;\overset{Fe^{2+};\;Fe^{3+}}{\to }O_{2}+OH^{-}+HO^{\cdot }$: Fenton/Haber–Weiss reaction

The toxicity of free iron has been extensively studied on neuronal and retinal cells, and they are not sensitive to the same doses of iron [46,92,93], the cones being the most sensitive to iron [94]. In RPE cells, the interaction of iron with bisretinoids and lipofuscin induces cell damage and retinal degeneration [91]. Conversely, melanin can bind large amounts of iron to preserve the RPE and the choroid from a pro-oxidant environment, intensified by light exposure. However, with age, the accumulation of iron in melanosomes associated with a reduction in the amount of melanin in RPE promotes the formation of free radicals [95]. Exposure of RPE cells to high non-lethal doses of iron leads to a decrease in phagocytic and lysosomal activity [15], favoring the accumulation of breakdown products of Vitamin A (lipofuscin) leading to the formation of glycation end products (AGE) present in drusen, RPE, and Bruch’s membrane of AMD patients [91]. In addition, phagocytosis of PR discs, peroxidized by ferrous ions, damage the membranes of phagosomes and lysosomes in RPE cells in culture [15,96].

In hypoxic conditions, an efflux of iron from RPE to the basolateral direction [97] could explain, at least in part, that PRs tolerate better hypoxia than hyperoxia [98]. Fe^2+^ contributes also to light-induced PR cell death through the production of hydroxyl radicals [99]. The ascorbate-Fe^2+^ complex induce lipid peroxidation in rod outer segment membranes and subsequently damage proteins such as rhodopsin by carbonylation or loss of thiol groups [100]. Finally, free heme can be also a source of redox-active iron and therefore highly toxic for the retina and for RPE cells [101].

In optic neuropathy, such as glaucoma, several mechanisms involved in ganglion cell death seem to be enhanced by iron-dependent oxidative stress [102,103].

Iron is thus a key component of oxidative-induced damages in the retina and in the RPE and involved in major cell death mechanisms.

#### 4.2.2. Retinal Cells Death Mechanisms in Iron Overload

Iron overload, induced experimentally by the implantation of iron particles in rat vitreous cavity caused apoptosis (TUNEL-positive nuclei) in the outer nuclear layer after only 2 days [104]. Rat retinal explant exposed to iron showed an early increase of necrotic markers, such as lactate deshydrogenase, receptor-interacting serine/threonine-protein (RIP) kinase, and incorporation of propidium iodide, even before intraretinal iron accumulation was detected. Using retinal organo-culture, it was observed that iron deposits in retinal explants induced a shift from necrosis to apoptosis with activation of caspase 3 and TUNEL-positive nuclei [105]. Increased intraocular iron levels following intravitreal FeSO_4_ injection caused oxidative damage of PR, as shown by the increase of superoxide radicals; hydroxynonenal, a marker of lipid peroxidation; and increased expression of heme oxygenase 1 [94]. Retinal iron overload also activates the NOD-like receptor family, pyrin domain containing 3 (NLRP3) inflammasome signaling pathway. In fact, the expression levels of NLRP3, activated caspase-1, a downstream target of NLRP3, and interleukin (IL) 1ß were higher in the retinas of HFE KO mice, a model of genetic iron overload [106]. Ferroptosis, a newly characterized form of necrosis, is induced by the accumulation of iron in degenerative diseases and has been described in RPE cells in culture subjected to oxidative stress [107]. Glutathione depletion also induced ferroptosis, autophagy, and premature senescence in RPE cells [108].

#### 4.2.3. Inflammation

The general implication of iron in inflammation has been recently reviewed, and it will not be detailed here [109]. In RPE cells, the intracellular accumulation of iron activated the NRLP3 inflammasome pathway via the repression of the degradation of aluRNA by double-stranded RNA-specific endoribonuclease (DICER1). This mechanism involved the sequestration of the cofactor PCBP2 [110] and has been advocated in AMD. It has been also reported that iron induces the synthesis of complement C3 by activation of the Extracellular signal-regulated kinases (ERK)/SMAD3/CCAAT Enhancer Binding Protein Delta (CEBPD) 48 pathway [111]. The complement factor C5 carries between 13 and 15 iron atoms necessary for its conversion into an active form C5b by C5 convertase, a complex formed from the cleavage products of C3 [112], which shows the importance of iron in complement pathways activation, a recognized risk factor for AMD [113].

The prion protein (PrP^C^), the principal protein implicated in the pathogenesis of human and animal prion disorders, is also implied in retinal degeneration due to iron metabolism dysfunction. This neuronal protein is expressed in many tissues of the eye, such as the retina and the cornea trabeculum. PrP^C^ is also expressed on the basolateral membrane of RPE, where it facilitates uptake of iron from choriocapillaries to neuroretina by functioning as a ferrireductase partner for divalent metal transporters. PrP-scrapie (PrP(Sc)), a misfolded isoform of this PrP^C^ accumulates in the neuroretina resulting in iron accumulation [114].

In the brain, IL6 produced by microglia in response to lipopolysaccharide, induced the production of HEPC by astrocytes [115]. HEPC prevented the iron overload-activated neuronal apoptosis [115]. LPS induced also an HFE-independent expression of HEPC in MGC and in the RPE, both in vitro and in vivo. The increase in HEPC levels in retinal cells, occurring with a decrease in FPN levels, led to oxidative stress and apoptosis within the retina in vivo [116]. On the other hand, in both in vitro and in vivo models of amyloid β-induced pathology, HEPC downregulates the inflammatory and pro-oxidant processes in astrocytes and microglia and protected neurons from cell death [117]. Microglia and MGC activation associated with reactive gliosis has been observed in HJV knockout mice (*Hjv*^−/−^ mice) with aging and subsequent retinal iron accumulation [37].

The role of LCN2 has been suspected in AMD where its expression is increased in aqueous humor and in the infiltrating cells present in the retina and choroid [118]. An age-related increase in LCN2 was described in RPE cells of Beta-crystallin A3 (Cryba1) conditional knockout mouse, a model of AMD associated with chronic inflammation response [49], but the exact implication of LCN2 in iron metabolism in these models remains to be studied.

#### 4.2.4. Angiogenesis

Increased iron levels in the retina could also have a role in the development of new vessels by inhibiting the anti-angiogenic effect of cleaved high molecular weight kininogen (Hka) [119], promoting the expression of succinate receptor 1 (SUCNR1 or GPR91) [120] which stimulates production of pro-angiogenic factors VEGF and angiopoietin [121]. *In Hjv*^−/−^ mice, that leads to abnormal retinal iron overload. Proliferation of new leaky blood vessels in the vitreous was associated with reactive gliosis involving MGC and microglia [37]. In addition to proliferation by migratory cells, intravitreal hemoglobin also stimulates a transient proliferation in cells of the RPE and possibly in some supportive cells of the neural retina, such as MGC and astrocytes [122]. Iron also plays a role in HIF transcriptional regulation of pro-angiogenic genes [51]

## 5. Role of Iron in Retinal Diseases

### 5.1. Siderosis and Retinal Hemorrhages

Eye siderosis is probably the first known manifestation of iron toxicity for the eye. The presence of a foreign body containing iron inside the eyeball leads to various clinical complications including heterochromia of the iris, mydriasis, cataract, and retinal and RPE atrophy. Electroretinography analysis shows a decrease in a and b wave amplitudes, due to the progressive degeneration of the cones and rods (for a review, see [123]). The increase in iron can be observed histologically as a granular structure with FT or hemosiderin into cells. The level of vitreous iron also increases [124].

Retinal hemorrhages are present in several retinal pathologies, such as exudative AMD, diabetic retinopathy, or myopic degeneration, and they are particularly deleterious for vision when located in the subretinal space. Vision loss is dependent on the size of the hemorrhage and the ability of the tissue to shed blood [125,126]. During sub-macular hemorrhage early PR damage has been reported within 24 h [127]. In rabbits, the injection of their own blood into the subretinal space leads to a progressive degeneration of the PRs from one day after injection until a total destruction at 7 days, with an accumulation of iron in the outer segments of PR and in RPE [128]. The increased release of iron from hemoglobin induces peroxidation of unsaturated phospholipids, which are extensively present in the retina and affects particularly retinal neurons compared to the retinal glial cells [129].

### 5.2. Retinal Manifestations of Inherited Iron Disorders

Iron can accumulate in the retina of patients with inherited diseases involving mutations in genes encoding proteins of iron homeostasis, which can cause an imbalance in the metabolism of retinal iron. The most common hereditary hemochromatosis is related to a mutation in the *Hfe* gene resulting in excessive absorption of iron by the intestine and its accumulation in the organs. Mutations in the *Tfr2*, *Fpn*, *Hjv*, and *Hepc* genes are also involved in the development of hemochromatosis. The clinical findings associated to the retinal iron accumulation in these patients, as well as the impact on visual function, are quite rarely reported due to the variability of penetrance and the existence of a treatment reducing systemic iron overload. However, iron deposits and other changes in the RPE as well as visual acuity loss have been already reported [130].

Aceruloplasminemia is an autosomal recessive disorder caused by mutations in the Cp gene, resulting in a defect in the export of iron from cells. The retina, brain, and pancreas are overloaded with iron, leading to the clinical consequences as retinal degeneration, dementia, and diabetes. Several cases associating yellow discoloration of the fundus, atrophy of the RPE, and drusen-like deposits in the macula have been described. In post-mortem sections, an accumulation of iron associated with an enlarged RPE and loss of pigment of the RPE was observed (for a review, see [27]).

In animal models invalidated for the genes coding for iron-related proteins, an accumulation of iron in the RPE and PR is systematically observed as well as abnormalities in the RPE and PR degeneration [25,35,39,52,131].

Studies carried out in aging rodents have shown that the increase in iron intakes in food or by intravenous injection leads to local iron deposits in the choroid, the RPE, and the segments of PR, as well as deposits of complement C3 in the Bruch’s membrane, hypertrophy, and vacuolation of RPE and changes in the choriocapillaries [132,133].

### 5.3. Age-Related Macular Degeneration

AMD is a leading cause of worldwide blindness in the elderly population, affecting 200 million individuals by 2020 and nearly 300 million by 2040 [134]. The pathological aging of the macula can cause dry or non-neovascular and wet or neovascular AMD. At the early stage, accumulation of extracellular material forms drusen between the basal lamina of the RPE and the inner layer of Bruch’s membrane in the eye. At the late stages, degeneration of the PR overlying the drusen can cause severe central vision loss in the dry form, whilst formation of new abnormal blood vessels from the choroid growing into the retina can cause subretinal fluid accumulation and bleeding. The wet form progresses rapidly and is responsible for 90% of severe vision loss associated with AMD. The pathogenesis of AMD is multifactorial, with genetic and environmental factors such as smoking. It is associated with dysregulations in the angiogenic, oxidative stress, lipid, inflammatory, and complement pathways [135]. Patients with early AMD have more iron in the macula than healthy patients. Iron deposits are found in the melanosomes of the choroid and the RPE, in the central layer of the calcified Bruch’s membrane, in the drusen, and at the level of the PR [136]. Part of this iron, found in the pathological retina of AMD patients, is in the toxic free form [137]. Patients with dry AMD have more than twice the concentration of iron in their aqueous humor than in patients with cataract surgery [138]. The macular region of AMD patients with geographic atrophy showed an increase in the expression of proteins involved in iron homeostasis such as TF, FT, and FPN in the PR layer and feet MGC [139]. TF and CP mRNAs are increased in the two advanced forms of AMD [140]. In the serum of patients with the different forms of AMD, a significant increase in TF and TFR1 and a significant decrease in the concentration of soluble FT were observed while iron levels were unchanged [141]. Several polymorphisms of the iron homeostasis genes have been associated with risk factors for AMD: *Tfr1*, *Tfr2* (obesity, tobacco) [142], *Dmt1* [143], *Irp1* and *Irp2* [143], and *heme oxygenases 1* and *2* (HO1/2) [144]. A recent study has shown that the expression of several miRNA, small non-coding RNA molecules binding in 3’UTR genes, was modified in the serum of AMD patients, especially those controlling the translation of the TFR1 and DMT1 proteins [145].

### 5.4. Diabetic Retinopathy

Diabetic retinopathy is a vision-threatening complication of diabetes affecting approximately 93 million in the middle-aged and elderly populations [146]. Chronic hyperglycemia causes progressive damage to retinal cells and to the retinal capillaries, leading to ischemia, VEGF-mediated retinal vascular abnormalization, and neovascular vessels that leak and bleed into the retina. Macular edema is also a major cause of vision loss in diabetic retinopathy [1]. Clinical reports have shown the link between iron levels in the vitreous and proliferative diabetic retinopathy [124,147]. A strong iron label was observed in the RPE and outer plexiform layer of patients with diabetic retinopathy [148]. In a mouse model of diabetic retinopathy, higher iron concentrations in the retina led to an increased expression of renin by a mechanism dependent on the GPR91 receptor [106].

### 5.5. Glaucoma Neuropathy

Glaucoma is increasingly a cause of irreversible blindness in the world. Its global prevalence is expected to be 76 million by 2020 and 112 million by 2040. Progressive damage to the optic nerve, leading to severe vision loss results from increased ocular pressure and other multiple favoring factors [149,150]. Although the link between iron and glaucoma is not yet fully understood, there is a change in iron homeostasis in glaucomatous eyes. TF concentration is increased in the aqueous humor [151], and mRNA of TF are increased in retina [152]. Whilst no differences were found in iron levels in aqueous humor of patients with primary open-angle glaucoma [153], serum levels of iron and FT were significantly increased [154,155], and serum CP level was lower [156]. A glaucomatous mice model had lower retinal iron concentrations than pre-glaucomatous DBA/2J and age-matched C57Bl/6J mice [157]. The expression of FT, CP, and TF was increased in monkey and rat glaucoma models [152,158]. In addition, the role of glutamate excitotoxicity in the pathogenesis of glaucoma is well documented; yet, there seems to be a link between the toxicity of glutamate and the increase in the entry of iron into neurons [159], and iron chelation seems to protect neurons against excitotoxicity and intraocular pressure-induced toxicity [160,161]. A mutation in the autophagy receptor optineurin is associated with the pathogenesis of glaucoma. It induces the degradation of the TFR1 and the Rab12-dependent autophagy mechanism leading to retinal ganglion cell death. The addition of iron in this model reduces cell death [162]. It seems that iron metabolism is dysregulated in glaucoma, but the exact role of iron is optic nerve damage and remains to be studied in the pathogenesis of glaucoma.

### 5.6. Inherited Retinal Dystrophies and Associated Diseases

Retinitis pigmentosa affects approximately 1.8 to 2.4 million people around the world. The disease is characterized by degeneration of the PR and progressive complete blindness [163]. Although iron has been shown to accumulate in several models of retinal degeneration, as in rd10 mouse or RCS rat [86,164], the direct link between iron and retinitis pigmentosa has not been established in human disease.

Macular telangiectasia type 2 (MacTel 2) is a complex macular disease, characterized by abnormal perifoveal vessels (telangiectasia), loss of retinal organization, and ultimately loss of macula function. MacTel2 is the only human disease recognized as primarily associated with MGC cells loss. It has been shown that iron accumulates in the retina of patients with MacTel 2. In a murine model of MGC ablation that mimics part of MacTel 2 phenotype, there is also an accumulation of iron in retina and in the RPE [148]. Knowing the importance of MGC cells in the regulation of iron levels in the retina, it could be hypothesized that iron accumulates in MacTel 2 as a consequence of MGC loss in the fovea [165].

## 6. Iron Neutralization as a Therapeutic Strategy for Retinal Diseases

### 6.1. Chemical Chelators

Whether iron dysmetabolism in the retina is a cause or a consequence of various retinal diseases, iron accumulation is pathogenic, and its neutralization was shown to protect the retina from oxidative damage and retinal cell death in various models using different neutralizing strategies [133] (Table 2). As early as in the 1970s, an iron chelator, Deferroxamine, was used in humans to reduce the amount of “rust” deposited on the eye with satisfactory results. Used in many other models of retinal degeneration (retinitis pigmentosa [166] or light-induced retinal damage models [167]), this chelator reduces the iron load and preserves the retina. Other chelators, such as Deferriprone, have shown significant protection of the retina in mice with impaired mechanisms of iron homeostasis [168,169,170,171]. These chemical chelators are mainly used clinically to treat hemosiderosis induced by frequent transfusions. Administered orally, subcutaneously or intramuscularly, they could led to several eye side effects, including vision loss [172,173]. These side effects could be explained because chemical iron chelators also bind the iron necessary for RPE and PR function [133,174].

As highlighted in a recent review [175], the clinical use of chemical chelators is complex because they should (1) target only the organ or tissue which is affected by the iron excess; (2) have a sufficient half-life; (3) cross the different barriers that surround the tissue; and (4) have a rapid elimination route.

### 6.2. Natural Chelators

Other natural molecules generally coming from plants, such as curcumin, polyphenols, and flavonoids, are iron chelators and have shown effectiveness in mouse models of retinal degeneration (for a review, see [186,187]).

### 6.3. Transferrin

TF is part of the TF superfamily, which also includes lactoferrin, melanotransferrin, and ovotransferrin, which are found in many species of both mammals and invertebrates. It consists in two lobes, each binding a Fe^3+^ atom with a very high affinity (10^22^M^−1^). Its primary role is to maintain an environment devoid of free iron. TF synthesized by RPE, PR, and neuronal cells is found in the aqueous and vitreous humors [8,105]. By single-cell RNA sequencing of human neural retina, mRNA for TF was enriched in peripheral retina compared to fovea [188]. Its expression is amplified during inflammation or immunity to increase the buffering capacity of iron. In light-induced retinal degeneration, TF and TFR1 mRNA increased in retina immediately after light exposure and then decreased at basal level. One day after light exposure, TF was increased, whereas TFR1 was reduced compared to not illuminated mice [32]. TF has long been of therapeutic interest due to its antimicrobial capacity and the ubiquitous presence of TFR1 allowing penetration of the blood-brain barrier [189]. TF has also been used successfully in humans in iron metabolism pathologies and for its cytoprotective capacity [190].

Our laboratory is interested in the potential of TF for the treatment of retinal pathologies (Table 3). Our work has shown that administration of the iron-free form (apoTF) by intraperitoneal injections in rd10 mice, a model of retinitis pigmentosa, preserves PRs better compared to the use of other chelators or antioxidants [46,164]. Injected into the vitreous, TF is present throughout the neural retina (MGC) and is eliminated via its receptors by RPE and the choroid without any immunogenic or toxic effect on the retina [46,105]. Thus, TF administered in a model of light-induced degeneration, allows the restoration of iron homeostasis, decreases iron accumulation, reduces inflammation and apoptosis, and preserves PRs and visual function [43]. In an ex vivo model of retinal detachment, TF inhibits the degenerative processes activated by the iron excess by reducing necrosis, apoptosis, gliosis, and oxidative stress. In vivo, human TF constitutively expressed in transgenic mice (TG) reduces loss of cones, cleavage of caspase 3, an apoptosis effector, DNA breaks, and necrosis (Figure 3). In rats, TF injected at the time of the detachment, reduces retinal edema, cell death and preserves PRs. In addition to its ability to reduce the accumulation of iron in the retina following detachment, TF also acts on other cellular pathways, no doubt through its interaction with molecular partners which remain to be discovered [105].

## 7. Conclusions

Iron is one of the most common elements on Earth. Two-hundred years ago, it was discovered that, after desiccation, the residual ashes of an aged human retina could be mobilized by a magnet (as quoted in [191]). Nowadays the chemical study of iron structure and its outer electrons has been revealed by the discovery of quantum effects of iron electron in biology. The best illustrations have been described by Cedric Weber who demonstrated that very specific quantum effects are involved to explain the energy in the binding of iron to oxygen and CO to hemoglobin [192]. This transition metal plays a main role in retinal physiology, but overload leads to retinal degeneration and loss of function. Iron chelation is a potential therapeutic target to prevent retinal degeneration. TF, as an endogenous iron binding protein, avoids toxic effects of iron depletion and activates additional neuroprotective pathways.

## Figures and Tables

**Figure 1 cells-09-00705-f001:**
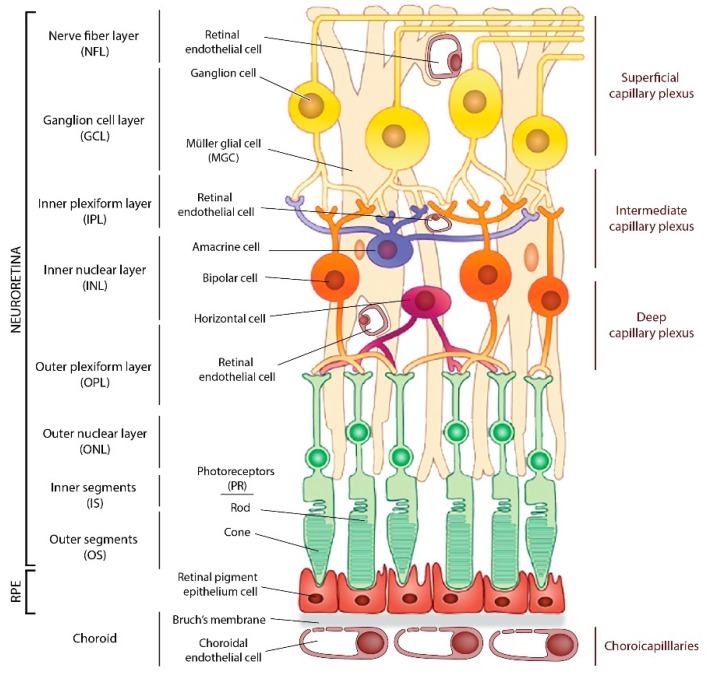
Schematic drawing of the cellular components of the retina. **Legend Figure 1:** There are three retinal vascular plexuses tightly coordinated with retinal neurons and a choroid plexus underlying RPE. GCL: Ganglion cell layer; NFL: Nerve fiber layer; INL: Inner nuclear layer; IPL: Inner plexiform layer; MGC: Müller glial cell; ONL: Outer nuclear layer; OPL: Outer plexiform layer; OS: Outer segments; IS: Inner segments; PR: Photoreceptors; RPE, retinal pigment epithelium.

**Figure 2 cells-09-00705-f002:**
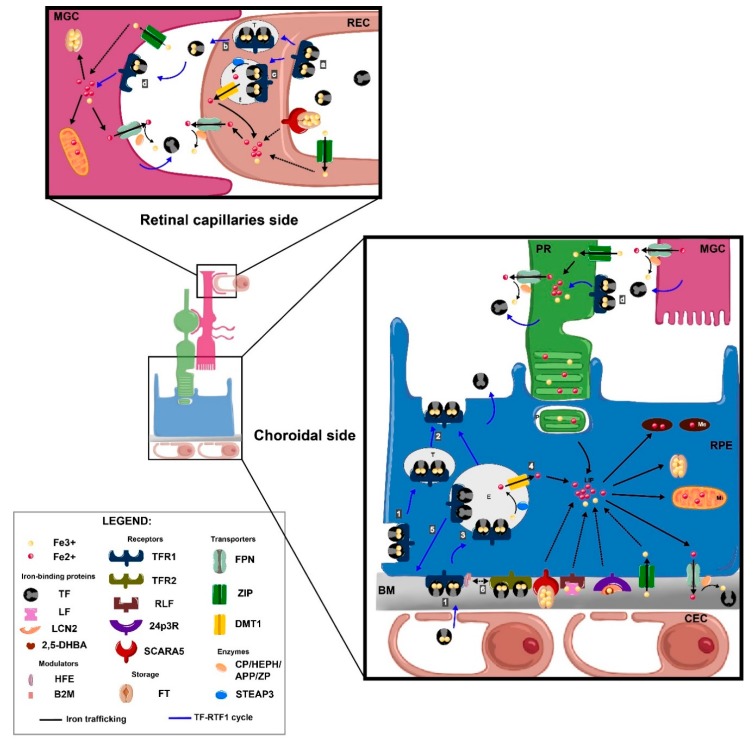
Iron uptake from capillaries and transport in the retina. **Legend Figure 2:** Under physiological condition, non-heme iron (Fe^3+^) in the circulation is transported bound to transferrin (TF). A. At the choroidal side, Fe^3+^ linked to TF is captured by its receptor 1 (TFR1) (1) at the basolateral level of the retinal pigment epithelium (RPE) (blue arrows). The internalized TF/TFR complex is transported to the apical pole by transcytosis (T) (2) or to the endosome (E) (3). In this case, Fe^3+^ is released from TF and reduced by the metaloreductase six transmembrane epithelial antigen of the prostate 3 (STEAP3) to ferrous iron (Fe^2+^) and then exported to the cytosol by the transporter divalent metal transporter 1 (DMT1) where it constitutes the free iron pool (LIP) (4). TF and TRF1 are recycled to membrane (5). Iron is then transported from LIP to the organelles as needed, either stored in ferritin (FT) and melanosomes (Me), or exported by ferroportin (FPN) coupled to ferroxidases such as ceruloplasmin (CP) or hephaestin (HEPH), amyloid-beta precursor protein (APP) or zyloklopen (ZP) (black arrows). Hemochromatosis protein (HFE) and beta-2 microglobulin (B2M) associated to TFR1, shift to TFR2 in case of iron overload (saturation of TF) and activate hepcidin (HEPC) transcription (6). B. At retinal capillaries side, Fe^3+^ bound to TF is up-taken by TFR1 at the luminal side of retinal endothelial cells (REC) (a), and TF/TFR1 pass directly through transcytosis into the retina (b) or endocytosed then exported by FPN (c). TF synthetized by RPE, Müller glial cells (MGC) or photoreceptors (PR) up-taken retinal iron (d) and distributed it throughout the retina, especially to PR. Phototransduction performed on the outer segments of PR is a highly iron-dependent process. PR uptake Fe^3+^ bound to TF by TFR1 presents in inner segments and export it by FPN or by phagocytosis (P) of the outer segments of PR by RPE. TF-independent iron delivery to the retina can occur, especially in case of systemic iron dysregulation (black dotted lines). Serum FT has a specific receptor, the scavenger receptor class A, member 5 (SCARA5) localizes at the basal membrane of RPE, luminal side of REC, PR and MGC. Lactoferrin (LF), a member of TF superfamily and its receptors (LFR) are present in RPE. Fe^3+^ captured by a siderophore (2,5-dihydroxybenzoic acid (2,5-DHBA)) is bound by lipocalin 2 (LCN2) and its receptors (24p3R) in RPE. The non-TF-bound iron (NTBI) is up-taken by MGC, REC, PR and RPE by DMT1 or ZRT/IRT-like proteins (ZIP) importers. BM: Bruch’s membrane; CEC: choroidal endothelial cell; E: endosome; Me: melanosome; MGC: Müller’s glial cell; MI: mitochondria; P: phagosome; PR: photoreceptor; REC: retinal endothelial cell; RPE: retinal pigment epithelium; T: transcytosis.

**Figure 3 cells-09-00705-f003:**
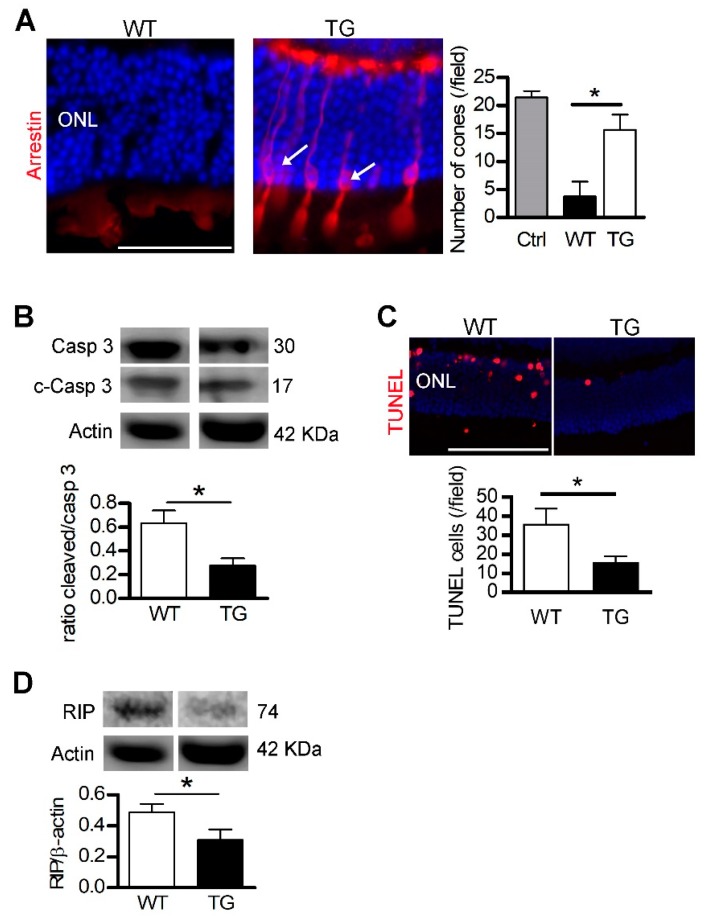
Transferrin expression preserves the detached retina. **Legend Figure 3:** After retinal detachment (RD), photoreceptors died by apoptosis and necrosis. Transgenic mice (TG) expressing human transferrin (TF) were used to demonstrate the protective effects of TF. (**A**) Arrestin staining revealed cones in retinal sections of TG mice (arrows) after RD. Cone number was higher in TG compared with WT mice. (**B**) The ratio of cleaved/pro–caspase 3 protein level was lower in TG mice compared to WT mice after RD. (**C**) The number of nuclei positive apoptotic-DNA breaks, stained by TUNEL, was reduced in TG mice compared to WT mice. (**D**) Necrotic RIP kinase protein level was reduced in TG mice compared with WT mice. All values are represented as the mean ± SEM. Mann–Whitney test (*n* = 3–6), * *p ≤* 0.025. ONL: Outer nuclear layer. Scale bar, 100 µm. From [105]. Reprinted with permission from AAAS.

**Table 1 cells-09-00705-t001:** Proteins involved in iron homeostasis of the retina.

	Proteins	Expression	Functions	Knock-Out Rodent Models	Human Pathologies
**Iron uptake/export**	Transferrin (TF)	RPE, PR, MGC [8]	Extracellular transporter binding two ferric iron ions (Fe^3+^) [holoTF]. Kd = 10^22^M^−1^	*Hpx*^−/−^: Decrease of the electroretinogram. Decrease TF, CP, TFR1 [17]	Congenital atransferrinemia
	Transferrin receptor 1 (TFR1)	RPE, IS, OPL, INL, GCL, endothelial cells [8]	Transmembrane receptor of holoTF	ND	ND
	Lactoferrin (LF)	RPE [18]	Extracellular transporter binding two Fe^3+^	*Lf*^−/−^: Higher susceptibility for laser induced choroidal neovascularization [19]	ND
	Lipocalin 2 (LCN2) or (neutrophil gelatinase-associated lipocalin (NGAL) or 24p3	RPE, MGC, neural retina, microglia [20]	Extracellular transporter which binds Fe^3+^ by sequestering bacterial and mammalian siderophores (2,5-dihydroxybenzoic acid).	*Lcn2*^−/−^: Expression of LFT, TF et TFR1 unchanged (personal data)	ND
	24p3R or (the solute carrier family 22 member 17 (SLC22A17)	RPE [21]	Transmembrane receptor of LCN2 under holo- and apo-forms		
	Megalin or (low density lipoprotein receptor-related protein 2 (LRP2)	RPE [22]	Transmembrane multiligands receptor (such lipocalin 2, lactoferrin, transferrin), co-receptor Cubulin	*Lrp2^F^*^/*F*^ (FoxG1^Cre+^): myopia, hypertrophic RPE and retinal degeneration [22]. Increase FT and decrease TFR1 (personal data)	Donnai-Barrow syndrome: high myopia, retinal detachment
	Ferroportin (FPN) or (SLC40A1)	RPE, IS, OPL, IPL, MGC, REC [23]	Transmembrane transporter which exports ferrous iron (Fe^2+^) outside the cell in cooperation with ferroxidases	*Fpn*^C326S^: HEPC-resistant FPN mice. Increase FT, iron deposits in RPE and choroid [24]	Hemochromatosis type 4
	Ceruloplasmin (CP)	RPE, MGC [25]	Extracellular ferroxidase that oxidizes Fe^2+^ in Fe^3+^. Exists a glycosylphosphatidylinositol-anchored form	*Cp*^−/−^*Heph*^sla/sla^: Neovessels, deposits under RPE, loss of RPE and PR. Iron accumulation in RPE and PR. Increase FT and decrease TFR1. *Cp*^−/−^*Heph*^F/F^ (Bestrophin1^Cre+^): no iron deposits and no retinal dystrophy [26]	Aceruloplasminemia: iron deposits in drusen and RPE. Dry-AMD like phenotype [27]
	Hephaestin (HEPH)	RPE, PR, MGC [25]	Extracellular ferroxidase. 50% of homology with CP		ND
	Amyloid-beta precursor protein (APP)	RPE, IS and OS, MGC, GCL [26]	Membrane ferroxidase	*App*^−/−^: Disturbances of synaptic development of secondary neurons [28].	Associated with Alzheimer disease and cerebral amyloid angiopathy
	Zyklopen (ZP)	RPE, GCL [26,29]	Membrane ferroxidase	ND	ND
	DMT1 (*Divalent Metal Transporter 1*) or Natural resistance-associated macrophage protein (NRAM2) or SLC11A2	IS, horizontal and rod bipolar cells [30]	Transmembrane import of Fe^2^ Iron exit from endosomal vesicle or cytosol under acidic pH (5.5).	ND	Microcytic hypochromic anemia with iron overload
	ZIP14 (*Zinc transporter 14*) or SLC39A14	CEC, RPE, PR, MGC, GCL, REC [31]	Transmembrane zinc transporter which uptakes unbound Fe^2+^ in cytosol. Optimal at physiological pH (7.4).	ND	Hypermanganesemia with dystonia 2; Hyperostosis cranialis interna.
	ZIP8 (*Zinc transporter 8*) or SLC39A8				Congenital disorder of glycosylation 2N
**Storage**	Ferritin (FT)	Ubiquitous and highly express in RPE, IS, bipolar cells [8]	Cytosolic complex of 24 subunits of heavy (H) and light (L) chains, which can store 4, 500 Fe^3+^. The H subunits have ferroxidase activity. FT has also a nuclear localization	*Hft*^+/−^: Higher sensibility for stress [32]	HFT: Hemochromatosis type 5LFT: Hyperferritinemia with or without cataract; Neuroferritinopathy; L-ferritin deficiency.
	Mitochondrial ferritin (FtMt)	All retina layers, with higher expression in RPE and ellipsoids of IS [30]	Mitochondria iron transporter. Share 79% of homology with HFT and has ferroxidase activity	ND	ND
**Regulation**	Transferrin receptor 2 (TFR2)	RPE, IS, OPL, IPL [33]	Transmembrane receptor of holo-TF which regulates transcription of HEPC in cooperation with HFE under TF iron-saturation	ND	Hemochromatosis type 3
	Hereditary hemochromatosis protein (HFE)	RPE [33]	Membrane protein which bind β2M to TFR1 or TFR2 in function of TF iron-saturation	*Hfe*^−/−^: Hypertrophy/Hyperplasia of RPE, PR degeneration. Increase FT [34]	Hemochromatosis type 1: Dysmorphism of RPE, drusen and alteration of vision [30]. Variegate porphyria. Microvascular complications of diabetes 7
	β-2-Microglobulin (β2M)	RPE, OS, IS, OPL, INL, IPL [33]	Membrane protein involved in HFE-TFR1/2 interaction	ND	Immunodeficiency 43; Amyloidosis 8.
	Bone Morphogenetic protein 6 (BMP6)	RPE, IS, OPL, IPL, GCL	Extracellular protein which regulates HEPC transcription. Bind Activin A receptor (Acvr1A) and BMP receptor type II (BMPR2) and HJV as coreceptor, all expressed in retina	*Bmp6*^−/−^: Iron accumulation in RPE and retina. RPE hypertrophy and PR degeneration. Decrease TFR1 and increase LFT [35]	ND
	Hemojuvelin (HJV)	RPE, PR, MGC, GCL [36]	Regulation of HEPC transcription	*Hjv*^−/−^: Neovessels in retina, gliosis, inner BRB leakage, PR degeneration. Increase LFT [37]	Hemochromatosis type 2A
	Transmembrane serine protease 6. (TMPRSS6) or Serine protease matriptase-2	RPE, MGC GCL [38]	Membrane protein with serine protease activity which cleave HVJ	Tmprss6^msk/msk^: No visual alteration [38]. Increase HEPC, HJV, TFR1	Iron-refractory iron deficiency anemia
	Hepcidin (HEPC)	RPE, IS, MGC, OPL [36]	Peptide hormone which transcription is activated by TF saturation or inflammation. Induces the degradation of FPN reducing iron export	*Hamp*^−/−^: Iron accumulation in RPE/choroid and in retina. Decrease TFR1 and increase FPN [39]	Hemochromatosis type 2B and juvenile
	Iron regulatory protein (IRP1) or cytoplasmic aconitase hydratase (ACO1)	Ubiquitous	Iron sensor protein with cluster iron-sulfur. Bind Iron responsive element (IRE) in target mRNA when intracellular iron levels are low. Under high iron condition, IRP1 is converted into an aconitase whereas IRP2 is degraded in proteasome	*Ireb1*^+/−^*Ireb2*^−/−^: No retinal alteration. Increase FPN and LFT [30]	ND
	IRP2 or Iron-responsive element-binding protein 2 (IREB2)				ND
	Hypoxia Inducible Factor (HIF)	RPE, PR ONL, INL, GCL [40,41]	Transcriptional regulator. Oxygen sensor sensitive to iron level. Bind Hypoxia responsive element (HRE) in target mRNA under hypoxia or when intracellular iron levels are low	ND	Familial erythrocytosis (HIF2α)

**Legend Table 1**: Proteins localization were obtained from immunostaining on sections of mouse/rat retinas. The mouse models presented are limited to those with retinal changes in iron homeostasis and retinal abnormalities, if any. The corresponding human diseases were obtained by searching the UniProt site. ND: Not determined. Legends: β2M: β-2-Microglobulin; CP: ceruloplasmin; CEC: choroidal endothelial cell; FT: ferritin; GCL: ganglion cells layer; HFE: hereditary hemochromatosis protein, HFT: ferritin heavy chain; IPL: inner plexiform layer; IRE: iron responsive element; IS: inner segments; LFT: ferritin light chain; MGC: Müller’s glial cell; OPL: outer plexiform layer; OS: outer segments; PR: photoreceptor; REC: retinal endothelial cell; RPE: retinal pigment epithelium; TFR1: transferrin receptor 1.

**Table 2 cells-09-00705-t002:** Comparation between chemical iron chelators and transferrin in clinical use.

	Deferoxamine	Deferiprone	Deferasirox	Transferrin
Iron Binding	1:1	3:1	2:1	2:1
Route of administration	Sub-cutaneous (every 8–12h) Intravenous (IV) (5 days/week)	Oral (t.i.d)	Oral (q.d)	Intravenous [176]
Half-Life(after IV administration)	20–30 min	3–4 h	8–16 h	4–8 d [176]
Excretion	Urinary/fecal	Urinary	Fecal	Unknown
Usual Doses (mg/Kg/d)	25–60 [177]	75–100 [177]	20–40 [177]	100 [176]
Clinical Use	Acute iron intoxicationChronic iron overload	Chronic iron overload	Chronic iron overload	Atransferrinemia [178] Haematological stem cell transplant [176]
Ocular Side effects	Pigmentary retinopathy [173], visual loss [179], impaired night vision [180], optic neuritis [173] and cataract [172].	Diplopia [181], cataract [182] and possible retinal toxicity [183].	Lens opacities [184] and retinal disorders [185]	No adverse effects observed

**Legend Table 2:** Tid: 3 times a day; q.d: once a day.

**Table 3 cells-09-00705-t003:** Transferrin as a therapeutic drug in retinal diseases models.

Model Experiment	Physiopathology	Administration Mode	Therapeutic Action of Transferrin	References
Primary culture of Müller glial cells.	Iron exposure	Cell isolation from transgenic mice carrying the human transferrin gene (TghTF)	Cell number preservation. Lower necrosis revealed by lactate dehydrogenase release. Inhibition of mRNA TF diminution.	[46]
Primary culture of Müller glial cells	Iron exposure	Addition of apo- or holo-human TF	Dose-dependent cell number preservation by apo- but not holo-human TF	[46]
rd10 mice	Model of retinitis pigmentosa presenting iron accumulation in photoreceptors (PR)	Crossing rd10 mice with TghTF mice	Preservation of retinal histology (outer and inner nuclear layers thickness). Less apoptotic-positive retinal cells.Conservation of rods and cones morphology	[164]
rd10 mice	Model of retinitis pigmentosa presenting iron accumulation in PR	Daily intraperitoneal injections of apo-human TF	Dose-dependent preservation of retinal histology (outer and inner nuclear layers thickness). Less apoptotic-positive retinal cells. Conservation of rods and cones morphology	[164]
Light-induced degeneration	Model of acute degenerative retina	Intravitreal injection of apo-human TF before and after light-induced degeneration	Preservation of retinal histology and functions. Preservation of ONL thickness and PR morphology. Lower ONL apoptotic- positive cells.Regulation of iron homeostasis balance. Lower retinal iron accumulation and oxidative stress. Regulation of retina inflammation and diminution of microglial cells activation in outer retina.	[43]
Light-induced degeneration	Model of acute degenerative retina	Electrotransfer of cDNA of human TF for *in oculo* production	Preservation of retinal histology and ONL layer thickness.	[43]
P23H rats	Model of retinitis pigmentosa	Electrotransfer of cDNA of human TF for *in oculo* production	Preservation of retinal histology and ONL layer thickness.	[43]
Bone morphogenetic protein 6 mice	Model of hemochromatosis with retinal iron accumulation	Intraperitoneal and intravitreal injections of apo-human TF	Diminution of iron accumulation in retina pigment epithelium	[43]
Retinal explant of mice	Retinal detachment with iron exposure	Retinas from TghTF	Preservation of cones number and rod outer segments length. Lower necrosis Prevention of iron retinal accumulation	[105]
Retinal explant of rats	Retinal detachment with iron exposure	Addition of apo-human TF after iron exposure	Preservation of rhodopsin expression level and cones number Lower necrosis and apoptosis Prevention of retinal iron accumulation	[105]
Subretinal injection of hyaluronic acid in mice	Retinal detachment presenting iron accumulation in subretinal space	TghTF mice	Preservation of retinal histology, rods outer segments length and number of cones Diminution of retinal oedema and Müller glial cells activation Lower apoptosis and necrosis Regulation of pathways involved in biological functions	[105]
Subretinal injection of hyaluronic acid in rats	Retinal detachment presenting iron accumulation in subretinal space	Intravitreal injection of apo-hTF	Preservation of retinal histology, rods outer segments length Diminution of retinal oedema	[105]

**Legend Table 3: ** ApoTF: transferrin without iron; HoloTF: transferrin binding iron; INL: inner nuclear layer; ONL: outer nuclear layer; PR: photoreceptors; TF: transferrin; TghTf: transgenic mice carrying the complete human transferrin gene.

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
