# Peer review of "From Rust to Quantum Biology: The Role of Iron in Retina Physiopathology"

_cells, 2020, doi:10.3390/cells9030705_

Round 1
Reviewer 1 Report
Overall, this is a comprehensive and helpful review of iron in retinal biology and toxicity. A few minor changes are needed:
Line: 18 change “endogen” to “endogenous’
Line: 88 change “RTF1” to “TFR1”. Also, change all subsequent references to transferrin receptor 1, which should be abbreviated “TFR1”. Transferrin receptor 2 should be abbreviated "TFR2"
Ln 249: The following statement is incorrect: “HIF-2α has an IRE sequence which in the condition of iron excess binds to IRP1 and causes its degradation.” The correct statement is “ HIF-2α has an IRE sequence in the 5’UTR of its mRNA, which, in the condition of iron deficiency, inhibits its translation”
In table 2, it is stated that deferiprone shows retinal toxicity, based on reference 198. This should be changed to “possible” retinal toxicity, as the RPE degeneration seen in patients with thalassemia on deferiprone occurred more frequently in those with the highest serum iron and ferritin levels. Thus, it may have been caused by iron overload rather than deferiprone. The paper compares the frequencies of retinal toxicity among those patients using deferiprone or not, but most patients in this study were on deferiprone, so the number not on deferiprone is too small for comparison.
Author Response
Thank you for your reviewing, found below the answers
- Line 18: “endogen” was replaced to “endogenous’.
- All “RTF1” were replaced to “TFR1”, “RTF2” was replaced to “TFR2”, and “RLF” was replaced by “LFR”. Changes were also made in table 1 and figure 2 and its legend
- Line 265: The sentence “HIF-2α has an IRE sequence in the 5’UTR of its mRNA, which, in the condition of iron deficiency, inhibits its translation” replaces previous one.
- Table 2: “possibly” was added in table 2
Reviewer 2 Report
Emilie Picard and colleagues describe the physiological and pathological roles of iron in the retina in this review article. Referring their own reports and others, they also mention the therapeutic effects of iron chelation in some retinal diseases. According to their expertise and great research experience, the literature is well summarized and important to be read. I have made some comments on the draft to be improved.
1. Line 52-53: Thus, most of the glucose is consumed in the outer retina through anaerobic glycolysis.
Please provide a reference to support this sentence.
2. In the Figure 1, abbreviations such as PR, MGC, and RPE may be shown and explained in the legend to easily understand those in the main text. Retinal endothelial cells existing in the inner plexiform layer may also be described as the intermediate capillary plexus. Inner and outer segments of photoreceptors should be indicated because their importance of iron transport and storage.
3. The Figure 2 is the most important presenting item for this article. However, the current scheme and the caption are a little confusing. The choroidal side and the retinal capillaries side can be divided to two different panels or figures. The sequence of iron transport may be described in the scheme with numbers and be explained in the legend.
4. Line 523: Although iron has been shown to accumulate in several models of retinal degeneration,
Please indicate specific animal models to accumulate iron with references.
5. Table I must be “Table 1”.
6. The reference 207 is a duplicate of the reference 43.
7. To show the retinal toxicity of iron and the therapeutic effect of transferrin, I suggest that the authors should present representative data as figures from the reference 114 with a journal permission.
Author Response
Thank you for your comments. Responses are below
1. Line 52-53: Thus, most of the glucose is consumed in the outer retina through anaerobic glycolysis.
Please provide a reference to support this sentence.
- Line 51: “Thus, most of the glucose is consumed in the outer retina through anaerobic glycolysis.” was deleted. We deleted theses sentences also : “The avascular nature of the outer retina explains that it is particularly vulnerable to changes in oxygen supply or demand. The oxygen tension varies as a function of retinal depth and geometry.”. We replaced them by “. In normal condition, the level of oxygen tension (Po2) in the outer retina is ten times lower than in the inner retina [3]. Oxygen and glucose consumption are metabolized to lactate, while aerobic glycolysis dominates energy production in the outer retina. Several factors modify Po2 level and utilization at the cellular level: the retinal depth, the light and hyperoxia [4,5]. PR have almost all mitochondria in their inner segments far from blood vessels. Light decreases oxygen utilization on the outer retina as much by a factor of two and increase Po2. Hyperoxia dramatically increases Po2 in the retina with the increase higher in outer retina compared to inner retina.” Two references were added [4,5]
2. In the Figure 1, abbreviations such as PR, MGC, and RPE may be shown and explained in the legend to easily understand those in the main text. Retinal endothelial cells existing in the inner plexiform layer may also be described as the intermediate capillary plexus. Inner and outer segments of photoreceptors should be indicated because their importance of iron transport and storage.
- Figure1 : Abbreviations were added in the scheme and in the legend. Retinal capillary plexus were added in the scheme. A legend was added.
3. The Figure 2 is the most important presenting item for this article. However, the current scheme and the caption are a little confusing. The choroidal side and the retinal capillaries side can be divided to two different panels or figures. The sequence of iron transport may be described in the scheme with numbers and be explained in the legend.
- Figure2: We changed title, caption and scheme. We added numbers and letters in the scheme to follow iron transport and added them in caption to easier comprehension. Choroidal side and retinal capillaries side were indicated on the scheme to help for differentiate the two compartments. Letters were added in the caption to help for comprehension. There are a new title and caption
4. Line 523: Although iron has been shown to accumulate in several models of retinal degeneration, Please indicate specific animal models to accumulate iron with references.
- Line 551: We changed the sentence: “Although iron has been shown to accumulate in several models of retinal degeneration, as in rd10 mouse or RCS rat, the direct link between iron and retinitis pigmentosa has not been established in human disease.” Two references were added [95, 180].
5. Table I must be “Table 1”.
- Table 1: Table I was renamed Table 1
6. The reference 207 is a duplicate of the reference 43.
- We deleted the reference 207
7. To show the retinal toxicity of iron and the therapeutic effect of transferrin, I suggest that the authors should present representative data as figures from the reference 114 with a journal permission.
- Line 609, we added “In vivo, human TF constitutively expressed in transgenic mice (TG) reduces loss of cones, apoptosis markers like DNA breaks and cleaved caspase 3, and necrosis protein (Figure 3). In rats, TF injected at the time of the detachment, reduces retinal edema, cell death and preserves PR.”
- Line 609, We added figure 3 with some data form the reference Daruich A et al. 2019. Journal’s permission to Science Advances has being requested.